# Accelerated Diffusion-Based Sampling by the Non-Reversible Dynamics with Skew-Symmetric Matrices

**DOI:** 10.3390/e23080993

**Published:** 2021-07-30

**Authors:** Futoshi Futami, Tomoharu Iwata, Naonori Ueda, Issei Sato

**Affiliations:** 1Communication Science Laboratories, NTT, Hikaridai, Seika-cho, “Keihanna Science City”, Kyoto 619-0237, Japan; tomoharu.iwata.gy@hco.ntt.co.jp (T.I.); naonori.ueda.fr@hco.ntt.co.jp (N.U.); 2Department of Computer Science, Graduate School of Information Science and Technology, The University of Tokyo, 7-3-1 Hongo, Bunkyo-ku, Tokyo 113-0033, Japan; sato@g.ecc.u-tokyo.ac.jp

**Keywords:** Markov Chain Monte Carlo, Langevin dynamics, Hamilton Monte Carlo, non-reversible dynamics

## Abstract

Langevin dynamics (LD) has been extensively studied theoretically and practically as a basic sampling technique. Recently, the incorporation of non-reversible dynamics into LD is attracting attention because it accelerates the mixing speed of LD. Popular choices for non-reversible dynamics include underdamped Langevin dynamics (ULD), which uses second-order dynamics and perturbations with skew-symmetric matrices. Although ULD has been widely used in practice, the application of skew acceleration is limited although it is expected to show superior performance theoretically. Current work lacks a theoretical understanding of issues that are important to practitioners, including the selection criteria for skew-symmetric matrices, quantitative evaluations of acceleration, and the large memory cost of storing skew matrices. In this study, we theoretically and numerically clarify these problems by analyzing acceleration focusing on how the skew-symmetric matrix perturbs the Hessian matrix of potential functions. We also present a practical algorithm that accelerates the standard LD and ULD, which uses novel memory-efficient skew-symmetric matrices under parallel-chain Monte Carlo settings.

## 1. Introduction

Sampling is one of the most widely used techniques for the approximation of posterior distribution in Bayesian inference [1]. Markov Chain Monte Carlo (MCMC) is widely used to obtain samples. In MCMC, Langevin dynamics (LD) is a popular choice for sampling from high-dimensional distributions. Each sample in LD moves toward a gradient direction with added Gaussian noise. LD efficiently explore around a mode of a target distribution using the gradient information without being trapped by local minima thanks to added Gaussian noise. Many previous studies theoretically and numerically proved LD’s superior performance [2,3,4,5]. Since non-reversible dynamics generally improves mixing performance [6,7], research on introducing non-reversible dynamics to LD for better sampling performance is attracting attention [8].

There are two widely known non-reversible dynamics for LD. One is underdamped Langevin dynamics (ULD) [9], which uses second-order dynamics. The other introduces perturbation, which consists of multiplying the skew-symmetric matrix by a gradient [8]. Here, we refer to the matrix as skew matrices for simplicity and this perturbation technique as skew acceleration. Much research has been done on ULD theoretically [9,10,11] and ULD is widely used in practice, which is also known as stochastic gradient Hamilton Monte Carlo [12]. In contrast, the application of the skew acceleration for standard Bayesian models is quite limited even though it is expected to show superior performance theoretically [8].

For example, skew acceleration has been analyzed focusing on sampling from Gaussian distributions [13,14,15,16,17], although assuming Gaussian distributions in Bayesian models is restrictive in practice. A recent study [8] theoretically showed that skew acceleration accelerates the dynamics around the local minima and saddle points for non-convex functions. Another work [18] clarified that the skew acceleration theoretically and numerically improves mixing speed when used as interactions between chains in parallel sampling schemes for non-convex Bayesian models.

Compared to ULD, what seems to be lacking for skew acceleration is a theoretical understanding of issues that are important to practitioners. The most significant problem is that no theory exists for selecting skew matrices. In existing studies, introducing a skew matrix into LD results in equal or faster convergence, denoting that a bad choice of skew matrix results in no acceleration. Thus, choosing appropriate skew matrices is critical. Furthermore, although ULD’s acceleration has been analyzed quantitatively, existing studies have only analyzed skew acceleration qualitatively. Thus, it is difficult to justify the usefulness of skew acceleration in practice compared to ULD. Another issue is that introducing skew matrices requires a vast memory cost in many practical Bayesian models.

The purpose of this study is to solve these problems from theoretical and numerical viewpoints and establish a practical algorithm for skew acceleration. The following are the two major contributions of this work.

Our contribution 1: We present a convergence analysis of skew acceleration for standard Bayesian model settings, including non-convex potential functions using Poincaré constants [19]. The major advantage of Poincaré constants is that we can analyze skew acceleration through a Hessian matrix and its eigenvalues and develop a practical theory about the selection of *J* and the quantitative assessment of skew acceleration.

Furthermore, we propose skew acceleration for ULD and present convergence analysis for the first time. Since ULD shows faster convergence than LD, combining skew acceleration with ULD is promising.

Our contribution 2: We develop a practical skew accelerated sampling algorithm for a parallel sampling setting with novel memory-efficient skew matrices. Since a naive implementation of skew acceleration requires a large memory cost to store skew matrices, memory-efficiency is critical in practice. We also present a non-asymptotic theoretical analysis for our algorithm in both LD and ULD settings under a stochastic gradient and Euler discretization. We clarify that introducing skew matrices accelerates the convergence of continuous dynamics, although it increases the discretization and stochastic gradient error. Then to the best of our knowledge, we propose the first algorithm that adaptively controls this trade-off using the empirical distribution of the parallel sampling scheme.

Finally, we verify our algorithm and theory in practical Bayesian problems and compare it with other sampling methods.

Notations: $I_{d}$ denotes a d×d identity matrix. Capital letters such as *X* represent random variables, and lowercase letters such as *x* represent non-random real values. ·, ∥·∥ and |·| denote Euclidean inner products, distances and absolute values.

## 2. Preliminaries

In this section, we briefly introduce the basic settings of LD and non-reversible dynamics for the posterior distribution sampling in Bayesian inference.

### 2.1. LD and Stochastic Gradient LD

First, we introduce the notations and the basic settings of LD and stochastic gradient LD (SGLD), which is a practical extension of LD. Here $z_{i}$ denotes a data point in space Z, |Z| denotes the total number of data points, and $x\in R^{d}$ corresponds to the parameters of a given model, which we want to sample. Our goal is to sample from the target distribution with density $d\pi \left(x\right)\propto e^{-\beta U(x)}dx$, where potential function U(x) is the summation of $u:R^{d}\times Z\to R$, i.e., $U\left(x\right)=\frac{1}{\left|Z\right|}\sum_{i=1}^{\left|Z\right|}u\left(x,z_{i}\right)$. Function u(·,·) is continuous and non-convex. The explicit assumptions made for it are discussed in Section 3.1. The SGLD algorithm [2,3] is given as a recursion:(1)$$\begin{array}{c} X_{k+1}=X_{k}-h\nabla \hat{U}\left(X_{k}\right)+\sqrt{2h\beta^{-1}}\epsilon_{k}, \end{array}$$
where $h\in R^{+}$ is a step size, $\epsilon_{k}\in R^{d}$ is a standard Gaussian random vector, β is a temperature parameter of π, and $\nabla \hat{U}\left(X_{k}\right)$ is a conditionally unbiased estimator of true gradient $\nabla U(X_{k})$. This unbiased estimate of the true gradient is suitable for large-scale data set since we can use not the full gradient, but a stochastic version obtained through a randomly chosen subset of data at each time step. This means that we can reduce the computational cost to calculate the gradient at each time step.

The discrete time Markov process in Equation (Equation 1) is the discretization of the continuous-time LD [2]:(2)$$\begin{array}{c} dX_{t}=-\nabla U\left(X_{t}\right)dt+\sqrt{2\beta^{-1}}dw_{t}, \end{array}$$
where $w_{t}$ denotes the standard Brownian motion in $R^{d}$. The stationary measure of Equation (Equation 2) is $d\pi \left(x\right)\propto e^{-\beta U(x)}dx$.

### 2.2. Poincaré Inequality and Convergence Speed

In sampling, we are interested in the convergence speed to the stationary measure. The speed is often characterized by the *the generator* associated with Equation (Equation 2) and defined as:(3)$$\begin{array}{cc} Lf(X_{t}): & =\lim_{s\to 0^{+}}\frac{E\left(f\left(X_{t+s}\right)|X_{t}\right)-f\left(X_{t}\right)}{s}\\ \; & =\left(-\nabla U\left(X_{t}\right)\cdot \nabla +\beta^{-1}\Delta \right)f\left(X_{t}\right), \end{array}$$
where Δ denotes a standard Laplacian on $R^{d}$ and f∈D(L) and $D\left(L\right)\subset L^{2}\left(\pi \right)$ denote the L domain. This −L is a self-adjoint operator, which has only discrete spectrums (eigenvalues). π with L has a *spectral gap* if the smallest eigenvalue of −L (other than 0) is positive. We refer to it as $\rho_{0}\left(>0\right)$. This spectral gap is closely related to Poincaré inequality. Internal energy is defined:(4)$$\begin{array}{c} E\left(f\right):=-\int_{R^{d}}fLfd\pi . \end{array}$$

Please note that E(f)>0 is satisfied. Then π with L satisfies the Poincaré inequality with constant *c*, if for any f∈D(L), π with L satisfies:(5)$$\begin{array}{c} \int f^{2}d\pi -{\left(\int fd\pi \right)}^{2}\leq cE\left(f\right). \end{array}$$

The spectral gap characterizes this constant $c\leq \frac{1}{\rho_{0}}$, which holds (see Section A.2 for details). We refer to best constant *c* as the Poincaré constant [19]. For notational simplicity, we define $m_{0}:=\frac{1}{c}$ and refer to this $m_{0}$ as the Poincaré constant.

In sampling, crucially, Poincaré inequality dominates the convergence speed in $\chi^{2}$ divergence:(6)$$\begin{array}{c} \int {\left(\frac{d\mu_{t}}{d\pi }-1\right)}^{2}d\pi :=\chi^{2}\left(\mu_{t}∥\pi \right)\leq e^{-\frac{2m_{0}}{\beta }t}\chi^{2}\left(\mu_{0}∥\pi \right), \end{array}$$
where $\mu_{t}$ denotes the measure at time *t* induced by Equation (Equation 2) and $\mu_{0}$ is the initial measure (see Section A.3 for details). Thus, the larger Poincaré constant $m_{0}$ is, the faster convergence we have.

### 2.3. Non-Reversible Dynamics

In this section, we introduce the non-reversible dynamics. π with L is reversible if for any test function f,g∈D(L), π with L satisfies
(7)$$\begin{array}{c} \int_{R^{d}}fLgd\pi =\int_{R^{d}}gLfd\pi . \end{array}$$

If this is not satisfied, π with L is non-reversible [19].

We introduce two non-reversible dynamics for LD. The first is ULD, which is given as
(8)$$\begin{array}{c} \begin{array}{cc} \; & dX_{t}=\Sigma^{-1}V_{t}dt,\\ \; & dV_{t}=-\nabla U\left(X_{t}\right)dt-\gamma \Sigma^{-1}V_{t}dt+\sqrt{2\gamma \beta^{-1}}dw_{t}, \end{array} \end{array}$$
where $V\in R^{d}$ is an auxiliary random variable, γ∈R is a positive constant, and Σ is the variance of the stationary distribution of auxiliary random variable *V*. The stationary distribution is $\tilde{\pi }:=\pi ⊗N\left(0,\Sigma \right)\propto e^{-\beta U\left(x\right)-\frac{1}{2}\Sigma^{-1}{∥v∥}^{2}}$, where N denotes a Gaussian distribution. The superior performance of ULD compared with LD has been studied rigorously [9,10,11]. ULD’s convergence speed is also characterized by the Poincaré constant [20]. In practice, we use discretization and the stochastic gradient for ULD, which is called the stochastic gradient Hamilton Monte Carlo (SGHMC) [10]. The second non-reversible dynamics is the skew acceleration given as
(9)$$\begin{array}{c} dX_{t}=-\left(I+\alpha J\right)\nabla U\left(X_{t}\right)dt+\sqrt{2\beta^{-1}}dw_{t}, \end{array}$$
where *J* is a real value skew matrix and $\alpha \in R^{+}$ is a positive constant. We call this dynamics S-LD. The stationary distribution of S-LD is still π, and S-LD shows faster convergence and smaller asymptotic variance [13,14,15,18].

## 3. Theoretical Analysis of Skew Acceleration

In this section, we present a theoretical analysis of skew acceleration in LD and ULD in standard Bayesian settings. We analyze acceleration through the Poincaré constant and connect it with the eigenvalues of the Hessian matrix, which allows us to obtain a practical criterion to choose skew matrices and quantitatively evaluate acceleration. We focus on a setting where a continuous SDE and a full gradient of the potential function is used in this section. The discretized SDE and stochastic gradient are discussed in Section 4.

### 3.1. Acceleration Characterization by the Poincaré Constant

First, we introduce the same four assumptions as a previous work [2], which showed the existence of the Poincaré constant about $m_{0}$ for LD (see Appendix C for details).

**Assumption 1.** 
*(Upper bound of the potential function at the origin) Function u takes nonnegative real values and is twice continuously differentiable on $R^{d}$, and constants A and B exist such that for all z∈Z,*
(10)$$\begin{array}{c} |u(0,z)|\leq A,\;∥\nabla u(0,z)∥\leq B. \end{array}$$


**Assumption 2.** 
*(Smoothness) Function u has Lipschitz continuous gradients; for all z∈Z, positive constant M exists for all $x,y\in R^{d}$,*
(11)$$\begin{array}{c} ∥\nabla u(x,z)-\nabla u(y,z)∥\leq M∥x-y∥. \end{array}$$


**Assumption 3.** 
*(Dissipative condition) Function u satisfies the (m,b)-dissipative condition for all z∈Z; for all $x\in R^{d}$, m>0 and b≥0 exist such that*
(12)$$\begin{array}{c} -x\cdot \nabla u\left(x,z\right)\leq -{m∥x∥}^{2}+b. \end{array}$$


**Assumption 4.** 
*(Initial condition) Initial probability distribution $\mu_{0}$ of $X_{0}$ has a bounded and strictly positive density $p_{0}$, and for all $x\in R^{d}$,*
(13)$$\begin{array}{c} \kappa_{0}:=\log\int_{R^{d}}e^{{∥x∥}^{2}}p_{0}\left(x\right)dx<\infty . \end{array}$$


Please note that these assumptions allow us to consider the non-convex potential functions, which are common in practical Bayesian models. Furthermore, we make the following assumption about *J*.

**Assumption** **5.** 
*The operator norm of J is bounded:*
(14)$$\begin{array}{c} {∥J∥}_{2}\leq 1. \end{array}$$
*This means that the largest singular value of J is below 1.*


Under these assumptions, we present the convergence behavior of skew acceleration using the Poincaré constant. First, we present the following S-LD result.

**Theorem** **1.**
*Under Assumptions 1–5, the S-LD of Equation (Equation 9) has exponential convergence,*
(15)$$\begin{array}{c} \chi^{2}\left(\mu_{t}^{\alpha }∥\pi \right)\leq e^{-\frac{2m(\alpha )}{\beta }t}\chi^{2}\left(\mu_{0}∥\pi \right), \end{array}$$
*where $\mu_{t}^{\alpha }$ is the measure at time t induced by S-LD and m(α) is the Poincaré constant of S-LD defined by its generator*
(16)$$\begin{array}{c} L_{\alpha }f\left(x\right):=\left(-\left(I+\alpha J\right)\nabla U\left(x\right)\cdot \nabla +\beta^{-1}\Delta \right)f\left(x\right). \end{array}$$
*Furthermore, m(α) satisfies $m\left(\alpha \right)\geq m_{0}$.*


The proof is shown in Appendix C. This theorem states that introducing the skew matrices accelerates the convergence of LD by improving the convergence rate from $m_{0}$ to m(α). Although [18] obtained a similar result, we used the Poincaré constant and derived an explicit criterion when $m\left(\alpha \right)=m_{0}$ holds, as we discuss below.

Next, we also introduce skew acceleration in ULD. Since ULD shows faster convergence than LD in standard Bayesian settings [10,11], it is promising to combine skew acceleration with ULD to obtain a more efficient sampling algorithm. For that purpose, we propose the following SDE:(17)$$\begin{array}{cc} \; & dX_{t}=\Sigma^{-1}V_{t}dt+\alpha_{1}J_{1}\nabla U\left(X_{t}\right)dt, \end{array}$$(18)$$\begin{array}{cc} \; & dV_{t}=-\nabla U\left(X_{t}\right)dt-\gamma \left(\Sigma^{-1}+\alpha_{2}J_{2}\right)V_{t}dt+\sqrt{2\gamma \beta^{-1}}dw_{t}, \end{array}$$
where $J_{1}$ and $J_{2}$ are real value skew matrices and $\alpha_{1}$ and $\alpha_{2}$ are positive constants. We assume that $J_{1}$ and $J_{2}$ satisfy Assumption 5. We refer to this method as skew underdamped Langevin dynamics (S-ULD) whose stationary distribution is $\tilde{\pi }=\pi ⊗N\left(0,\Sigma \right)\propto e^{-\beta U\left(x\right)-\frac{1}{2}\Sigma^{-1}{∥v∥}^{2}}$. See Appendix B for details, which include discussions on other combinations of skew matrices. As for S-ULD, we need an additional assumption about the initial condition of $V_{0}$:

**Assumption 6.** 
*(Initial condition) Initial probability distribution $\mu_{0}\left(x,v\right)$ of $\left(X_{0},V_{0}\right)$ has a bounded and strictly positive density $p_{0}$ that satisfies,*
(19)$$\begin{array}{c} \kappa_{0}:=\log\int_{R^{2d}}e^{{∥x∥}^{2}+{∥v∥}^{2}}p_{0}\left(\left(x,v\right)\right)dxdv<\infty . \end{array}$$


We then provide the following convergence theorem that resembles S-LD.

**Theorem** **2.**
*Under Assumptions 1–3, 5, 6, S-ULD has exponential convergence in $\chi^{2}$ divergence and its convergence rate is also characterized by m(α) as defined in Theorem 1. S-ULD’s convergence equals or exceeds ULD, of which convergence rate is characterized by $m_{0}$.*


See Section C.2 for details. From these theorems, we confirmed that skew acceleration is effective in both S-LD and S-ULD, and the convergence speed is characterized by Poincaré constant m(α) defined by Equation (Equation 16).

### 3.2. Skew Acceleration from the Hessian Matrix

Our goal is to clarify what choices of *J* induce $m\left(\alpha \right)>m_{0}$, which leads to acceleration. Therefore, we discuss how Poincaré constant m(α) is connected to the eigenvalues and eigenvectors of the perturbed Hessian matrix $\left(I+\alpha J\right)\nabla^{2}U\left(x\right)$. Next, we introduce the notations. We express the Hessian of U(x) as H(x) and the perturbed Hessian matrix as $H^{\prime }\left(x\right):=\left(I+\alpha J\right)H\left(x\right)$. Please note that *H* is a real symmetric matrix, which has real eigenvalues and diagonalizable. On the other hand, since $H^{\prime }$ is not symmetric, it has complex eigenvalues, although diagonalization is not assured (see Appendix E). We express pairs of eigenvectors and eigenvalues of $H^{\prime }\left(x\right)$ as ${\left\{\left(v_{i}^{\alpha }\left(x\right),\lambda_{i}^{\alpha }\left(x\right)\right)\right\}}_{i=1}^{d}$, which are ordered as Re(λ1α(x)))≤⋯≤Re(λdα(x)). Here, Re(λ1α(x)) expresses the real part of complex value $\lambda_{1}^{\alpha }$ and Im denotes the imaginary part. We express those of H(x) as ${\left\{\left(v_{i}^{0}\left(x\right),\lambda_{i}^{0}\left(x\right)\right)\right\}}_{i=1}^{d}$ and order them as $\lambda_{1}^{0}\left(x\right)\leq \cdots \leq \lambda_{d}^{0}\left(x\right)$.

#### 3.2.1. Strongly Convex Potential Function

Assume that *U* is an m-strongly convex function, where for all $x\in R^{d}$, $m\leq \lambda_{1}^{0}\left(x\right)$ holds. Poincaré constant $m_{0}$ of LD satisfies $m_{0}=m$ [19]. For the skew acceleration, since Poincaré constant satisfies $m\left(\alpha \right)=m^{\prime }\left(\alpha \right)$, where $m^{\prime }\left(\alpha \right)$ is the best constant that satisfies, for all *x*, m′(α)≤Reλ1α(x) (see Section D.1). Therefore, studying the Poincaré constant is equivalent to studying the smallest (real part of the) eigenvalue of the Hessian matrix. Thus, the relation between $\lambda_{1}^{0}\left(x\right)$ and Reλ1α(x) must be studied. The following theorem describes how the skew matrices change the smallest eigenvalue.

**Theorem** **3.**
*For all $x\in R^{d}$, the real parts of the eigenvalues of $H^{\prime }$ satisfy*
(20)m≤λ10(x)≤Reλ1α(x)≤⋯≤Reλdα(x)≤λd0(x).
*The condition of λ10(x)=Reλ1αx)) is shown in Remark 1.*


**Remark** **1.***Denote the set of the eigenvectors of eigenvalue $\lambda_{1}^{0}\left(x\right)$ as $V_{1}^{0}$. If $V_{1}^{0}=\left\{v\right\}$ and Jv=0, then λ10(x)=Reλ1αx)) holds. If the cardinality of set $V_{1}^{0}$ is larger than* 1, *and vectors $v,v^{\prime }\in V_{1}^{0}$ exist, such that λ10αJv=(Imλ1α)v′ and λ10αJv′=−(Imλ1α)v, then λ10(x)=Reλ1αx)) holds.*

Refer to Appendix F for the proof. This is an extension of previous work [8,13]. If λ10(x)<Reλ1α(x) is satisfied for all *x*, we have $m_{0}<m\left(\alpha \right)$, i.e., acceleration occurs. We discuss how to construct *J* such that λ10(x)<Reλ1α(x) holds in Section 3.3.

#### 3.2.2. Non-Convex Potential Function

The previous work [21] clarified that the Poincaré constant of the non-convex function is characterized by the negative eigenvalue of the saddle point. As shown in Figure 1, denote $x_{1}$ as the global minima, and $x_{2}$ is the local minima which has the second smallest value in U(x). We express the saddle point with index one, i.e., there is only one negative eigenvalue at the point, between $x_{1}$ and $x_{2}$ as $x^{*}$. This means that the eigenvalues of $H(x^{*})$ satisfies $\lambda_{1}^{0}\left(x^{*}\right)<0<\lambda_{2}^{0}\left(x^{*}\right)<\cdots <\lambda_{d}^{0}\left(x^{*}\right)$. Ref. [21] clarified that the saddle point $x^{*}$ characterizes the Poincaré constant as
(21)$$\begin{array}{c} m_{0}^{-1}\propto \frac{1}{|\lambda_{1}\left(x^{*}\right)|}e^{\beta (U\left(x^{*}\right)-U\left(x_{1}\right)-U\left(x_{2}\right))}. \end{array}$$
When skew matrices are introduced, [8] clarified the following relation:

**Theorem 4.** 
*([8]) $\lambda_{1}^{\alpha }\left(x^{*}\right)\leq \lambda_{1}^{0}\left(x^{*}\right)<0$ and equality holds only if $Jv_{1}^{\alpha }\left(x^{*}\right)=0$.*


Note $\lambda_{1}^{\alpha }\left(x^{*}\right)$ is not a complex number. Thus, the skew acceleration reduces the negative eigenvalue and leads to a larger Poincaré constant (see Section D.2) and results in faster convergence.

In conclusion, introducing the skew matrix changes the Hessian’s eigenvalues and increase the Poincaré constant. If λ10(x)≠Reλ1α(x) is satisfied, this leads to faster convergence for both convex and non-convex potential functions.

### 3.3. Choosing J

In this section, we present a method for choosing *J* that leads to λ10(x)≠Reλ1α(x) to ensure the acceleration based on the equality conditions in Theorems 3 and 4. Combining these theorems, we obtain the following criterion:

**Remark** **2.**
*Given a point x, λ10(x)≠Reλ1α(x) holds if either the following conditions are satisfied: (i) when $V_{1}^{0}=\left\{v\right\}$, Jv≠0 is satisfied. (ii) when $|V_{1}^{0}|>1$, Jv≠0 holds for any $v\in V_{1}^{0}$, and for any $v,v^{\prime }\in V_{1}^{0}$, λ10αJv=(Imλ1α)v′ and λ10αJv′=−(Imλ1α)v are not satisfied.*


The first condition (i) is easily satisfied if we choose *J* such that KerJ={0}. On the other hand, the second condition (ii) is difficult to verify since *H* and its eigenvalues and eigenvectors generally depend on the current position of $X_{t}$. Instead of evaluating eigenvalues and eigenvectors of *H* and $H^{\prime }$ directly, we use the random matrix property shown in the next theorem.

**Theorem** **5.**
*Suppose the upper triangular entries of J follow a probability distribution that is absolutely continuous with respect to the Lebesgue measure. If KerJ={0} is satisfied, then given a point $x\in R^{d}$, λ10(x)≠Reλ1α(x) holds with probability 1.*


The proof is given in Section G.1. From this theorem, we simply generate *J* from some probability distribution, such as the Gaussian distribution. Then, we check whether KerJ={0} holds. If KerJ={0} does not hold, we generate a random matrix *J* again.

The above theorem is valid only at a given evaluation point *x*. We can extend the above theorem to all the points over the path of the discretized dynamics (see Section G.3). With this procedure, we can theoretically ensure that acceleration occurs with probability one for discretized dynamics.

### 3.4. Qualitative Evaluation of The Acceleration

So far, we have discussed skew acceleration qualitatively but not quantitatively. Although acceleration’s quantitative evaluation is critical for practical purposes, to the best of our knowledge, no existing work has addressed it. In this section, we present a formula that quantitatively assesses skew acceleration by analyzing the eigenvalues of the Hessian matrix.

**Theorem** **6.**
*With the identical notation as in Theorem 3, for all x, we have*
(22)Reλ1α(x)=λ10(x)+α2∑k=2dλ10(x)λk0(x)|vk0(x)Jv10(x)|2λk0(x)−λ10(x)+O(α3).

*In particular, at saddle point $x^{*}$, we have*
(23)$$\begin{array}{c} \lambda_{1}^{\alpha }\left(x^{*}\right)=\lambda_{1}^{0}\left(x^{*}\right)+\alpha^{2}\sum_{k=2}^{d}\frac{\lambda_{1}^{0}\left(x^{*}\right)\lambda_{k}^{0}\left(x^{*}\right){\left|v_{k}^{0}\left(x^{*}\right)Jv_{1}^{0}\left(x^{*}\right)\right|}^{2}}{\lambda_{k}^{0}\left(x^{*}\right)-\lambda_{1}^{0}\left(x^{*}\right)}+O\left(\alpha^{3}\right). \end{array}$$


The proofs are shown in Appendix H. When focusing on Equation (Equation 22), if U(x) is a strongly convex function, since for all k>1, $\lambda_{k}\left(x\right)>\lambda_{1}\left(x\right)>0$ holds and the second term in Equation (Equation 22) is positive. From this, Reλ1α(x)>λ10(x) holds. A similar relation holds for Re(λdα(x)). In Equation (Equation 23), $\lambda_{1}^{\alpha }\left(x^{*}\right)<\lambda_{1}^{0}\left(x^{*}\right)<0$ holds. Thus, the changes of the Poincaré constants are proportional to $\alpha^{2}$. With these formulas, we can quantitatively evaluate the acceleration. We present numerical experiments to confirm our theoretical findings in Section 6.1.

## 4. Practical Algorithm for Skew Acceleration

In this section, we discuss skew acceleration in more practical settings compared to Section 3. First, we discuss the memory issue for storing *J* and the discretization of SDE and the stochastic gradient, which are widely used techniques in Bayesian inference. Finally, we present a practical algorithm for skew acceleration.

### 4.1. Memory Issue of Skew Acceleration and Ensemble Sampling

For *d*-dimensional Bayesian models, we need $O(d^{2})$ memory space to store skew matrices *J*s, and this is difficult for high-dimensional models. Instead of storing *J*, we can randomly generate *J*s at each time step following Theorem 5. However, we experimentally confirmed that using different *J*s at each step does not accelerate the convergence (see Section 6). Thus, we need to use a fixed *J* during the iterations.

As discussed below, we found that the previously proposed accelerated parallel sampling [18] can be a practical algorithm to resolve this memory issue. In that method, we simultaneously updated *N* samples of the model’s parameters with correlation. In such a parallel sampling scheme, a correlation exists among multiple Markov chains, it is more efficient than a naive parallel-chain MCMC, where the samples are independent. We express the *n*-th sample at time *t* as $X_{t}^{\left(n\right)}\in R^{d}$ and the joint state of all samples at time *t* as $X_{t}^{⊗N}:={\left(X_{t}^{\left(1\right)},\ldots ,X_{t}^{\left(N\right)}\right)}^{⊤}\in R^{dN}$. We express the joint stationary measure as $\pi^{⊗N}:=\pi ⊗\cdots ⊗\pi \left(x^{⊗N}\right)\propto e^{-\beta \sum_{i=1}^{N}U\left(x^{\left(i\right)}\right)}$. We express the sum of the potential function as $U^{⊗N}:=\sum_{i=1}^{N}U\left(x^{\left(i\right)}\right)$. We then consider the following dynamics:(24)$$\begin{array}{cc} \; & dX_{t}^{⊗N}\;=\;-\left(I_{dN}+\alpha J\right)\nabla U^{⊗N}\;\left(X_{t}^{⊗N}\;\right)dt\;+\;\sqrt{2\beta^{-1}}dw_{t},\;\;\; \end{array}$$(25)$$\begin{array}{cc} \; & \nabla U^{⊗N}\left(X_{t}^{⊗N}\right):={\left(\nabla U\left(X_{t}^{\left(1\right)}\right),\ldots ,\nabla U\left(X_{t}^{\left(N\right)}\right)\right)}^{⊤}. \end{array}$$
We call this dynamics skew parallel LD (S-PLD). *N*-independent parallel LD (PLD) is coupled with the skew matrix. Since each chain in PLD is independent of the other, the Poincaré constant of PLD is also $m_{0}$. Ref. [18] argued that the Poincaré constant of S-PLD, m(α,N), satisfies $m\left(\alpha ,N\right)\geq m_{0}$. This means S-PLD shows faster convergence than PLD. As discussed in Section 3.2, these Poincaré constants are characterized by the smallest eigenvalue of the Hessian matrix $\nabla^{2}U^{⊗N}\left(x^{⊗N}\right)$ and $\left(I_{dN}+\alpha J\right)\nabla^{2}U^{⊗N}\left(x^{⊗N}\right)$ where $x^{⊗N}\in R^{dN}$. We denote these smallest eigenvalues as $\lambda_{1}^{0}\left(x^{⊗N}\right)$ and Reλ1α(x⊗N). As discussed in Section 3.2, acceleration occurs if λ10(x⊗N)≠Reλ1α(x⊗N) is satisfied.

In [18], they failed to specify the choice of *J* whose naive construction of *J* requires $O(d^{2}N^{2})$ memory cost. To reduce the memory cost, we propose the following skew matrix:(26)$$\begin{array}{c} J:=J_{0}⊗I_{d}, \end{array}$$
where $J_{0}$ is a N×N skew matrix and ⊗ is a Kronecker product. We then have the following lemma:

**Lemma** **1.**
*If $J_{0}$ is generated based on Theorem 5 and $\mathrm{Ker}J_{0}=\left\{0\right\}$ is satisfied, then given a point $x^{⊗N}$, J does not satisfy the equality condition in Theorems 3,4, which means λ10(x⊗N)≠Reλ1α(x⊗N) with probability 1.*


See Section G.2 for the proof. Thus, from this lemma, we only need to prepare and store $J_{0}$, which requires $O(N^{2})$ memory, which does not depend on *d*. In practical settings, this is a significant reduction of the memory size since the number of parallel chains is smaller than the dimension of models. Please note that we can ensure the acceleration with this *J*.

**Lemma** **2.**
*Under Assumptions 1–5, assume J satisfies the condition of Lemma 1. Then S-PLD shows*
(27)$$\begin{array}{c} \chi^{2}\left(\mu_{t}^{\alpha ,⊗N}∥\pi^{⊗N}\right)\leq e^{-\frac{2m(\alpha ,N)}{\beta }t}\chi^{2}\left(\mu_{0}^{⊗N}∥\pi^{⊗N}\right), \end{array}$$
*where $\mu_{t}^{\alpha ,⊗N}$ is the measure at time t induced by S-PLD, and $\mu_{0}^{⊗N}$ is the initial measure defined as the product measure of $\mu_{0}$.*


See Section I.1 for the proofs. Thus, combined with Lemma 2, S-PLD converges faster than PLD. We also considered the ensemble version of ULD (parallel ULD (PULD)) and its skew accelerated version:(28)$$\begin{array}{c} \begin{array}{cc} \; & dX_{t}^{⊗N}=\Sigma^{-1}V_{t}^{⊗N}dt+\alpha_{1}J_{1}\nabla U^{⊗N}\left(X_{t}^{⊗N}\right)dt,\\ \; & dV_{t}^{⊗N}=-\nabla U^{⊗N}\left(X_{t}^{⊗N}\right)dt-\gamma \left(\Sigma^{-1}+\alpha_{2}J_{2}\right)V_{t}^{⊗N}dt+\sqrt{2\gamma \beta^{-1}}dw_{t}, \end{array} \end{array}$$
where $J_{1}$ and $J_{2}\in R^{dN\times dN}$ are real-valued skew-symmetric matrices, and $\alpha_{1}$ and $\alpha_{2}\in R_{+}$ are positive constants and $V_{t}^{⊗N}={\left(V_{t}^{\left(1\right)},\ldots ,V_{t}^{\left(N\right)}\right)}^{⊤}\in R^{dN}$. We refer to this dynamics as skew PULD (S-PULD) whose faster convergence can be assured similar to Lemma 2 as shown in Section I.2.

### 4.2. Discussion of the Discretization of SDE and Stochastic Gradient and Practical Algorithm

In this section, we further consider practical settings for S-PLD and S-PULD. We discretize these continuous dynamics, e.g., by the Euler-Maruyama method, and approximate the gradient by the stochastic gradient. Although introducing skew matrices accelerates the convergence of continuous dynamics, it simultaneously increases the discretization and stochastic gradient error, resulting in a trade-off. We present a practical algorithm that controls this trade-off.

#### 4.2.1. Trade-off Caused by Discretization and Stochastic Gradient

We consider the following discretization and stochastic gradient for S-PLD and S-PULD:(29)$$\begin{array}{c} X_{k+1}^{⊗N}=X_{k}^{⊗N}-h\left(I_{dN}+\alpha J\right)\nabla {\hat{U}}^{⊗N}\;\left(X_{k}^{⊗N}\;\right)+\sqrt{2h\beta^{-1}}\epsilon_{k}, \end{array}$$
and
(30)$$\begin{array}{c} \begin{array}{cc} \; & X_{k+1}^{⊗N}=X_{k}^{⊗N}+\Sigma^{-1}V_{k}^{⊗N}h+\alpha J\nabla {\hat{U}}^{⊗N}\left(X_{k}^{⊗N}\right)h\\ \; & V_{k+1}^{⊗N}=V_{k}^{⊗N}\;-\;\nabla {\hat{U}}^{⊗N}\left(X_{k}^{⊗N}\right)h\;-\;\gamma \Sigma^{-1}V_{k}^{⊗N}h\;+\;\sqrt{2\gamma \beta^{-1}h}\epsilon_{k}, \end{array} \end{array}$$
where $\epsilon_{k}\in R^{dN}$ is a standard Gaussian random vector. $\nabla {\hat{U}}^{⊗N}\;\left(X^{⊗N}\;\right)$ is an unbiased estimator of the gradient $\nabla U^{⊗N}\;\left(X^{⊗N}\;\right)$. We refer to Equation (Equation 29) as skew-SGLD and Equation (Equation 30) as skew-SGHMC. For skew-SGHMC, we dropped $J_{2}$ of S-PULD to decrease the parameters, shown in Appendix B. Please note that skew-SGLD is the identical as the previous dynamics [18]. We introduce an assumption about the stochastic gradient:

**Assumption 7.** 
*(Stochastic gradient) There exists a constant δ∈[0,1) such that*
(31)$$\begin{array}{c} E[∥\nabla \hat{U}\left(x\right)-\nabla U\left(x\right){∥}^{2}]\leq 2\delta \left(M^{2}{∥x∥}^{2}+B^{2}\right). \end{array}$$


Given a test function *f* with $L_{f}$ lipschitzness, we approximate ∫fdπ by skew-SGLD or skew-SGHMC, with estimator $\frac{1}{N}\sum_{n=1}^{N}f\left(X_{k}^{\left(n\right)}\right)$. The bias of skew-SGLD is upper-bounded as

**Theorem** **7.**
*Under Assumptions 1–7, for any k∈N and any $h\in (0,1\wedge \frac{m}{4M^{2}})$ obeying kh≥1 and βm≥2, we have*
(32)$$\begin{array}{cc} \; & \left|E\frac{1}{N}\sum_{n=1}^{N}f\left(X_{k}^{\left(n\right)}\right)-\int_{R^{d}}fd\pi \right|\leq L_{f}\left(\underset{\left(i\right)}{\underbrace{C_{1}\left(\alpha \right)kh}}+\underset{\left(ii\right)}{\underbrace{C_{2}e^{-\beta^{-1}m\left(\alpha ,N\right)kh}}}\right) \end{array}$$
*and $C_{1}$ and $C_{2}$ depends on the constants of Assumptions 1–7, for the details see Appendix J.*


We present a tighter bias bound in Section 4.3 under a stronger assumption. We can show a similar upper bound for the skew-SGHMC using the same proof strategy. This bound resembles of a previous one [18]; ours shows improved dependency on kh. The previous results of [18] are also limited to LD, not including skew-SGHMC.

Please note that (i) corresponds to the discretization and stochastic gradient error and (ii) corresponds to the convergence behavior of S-PLD, which is continuous dynamics. Since $C_{1}\left(\alpha \right)\geq C_{1}\left(\alpha =0\right)$, skew acceleration increases the discretization and stochastic gradient error. On the other hand, since $m\left(\alpha ,N\right)\geq m_{0}$, the convergence of the continuous dynamics is accelerated. Thus, skew acceleration causes a trade-off. When α is sufficiently small, we derive the explicit dependency of α for this trade-off from an asymptotic expansion. Using the quantitative evaluation of skew acceleration in Theorem 6, we obtain
(33)$$\begin{array}{cc} \; & \left|E\frac{1}{N}\sum_{n=1}^{N}f\left(X_{k}^{\left(n\right)}\right)-\int_{R^{d}}fd\pi \right|\leq \underset{\left(i\right)}{\underbrace{(d_{1}\alpha +d_{2}\alpha^{2})kh}}-\underset{\left(ii\right)}{\underbrace{\alpha^{2}d_{0}e^{-\beta^{-1}m_{0}kh}}}+O\left(\alpha^{3}\right)+\mathrm{const}, \end{array}$$
where $d_{0}$ to $d_{2}$ are positive constants obtained by the asymptotic expansion. See Appendix K for the details. In the above expression, (i) and (ii) correspond to (i) and (ii) of Equation (Equation 32). Thus, by choosing appropriate α, we can control the trade-off.

#### 4.2.2. Practical Algorithm Controlling the Trade-off

Since calculating the optimal α that minimizes Equation (Equation 33) at each step is computationally demanding, we adaptively tune the value of α by measuring the acceleration with kernelized Stein discrepancy (KSD) [22]. Our idea is to update samples under different α and α+η, and compare KSD between the stationary and empirical distributions of these different interaction strengths. Here, $\eta \in R^{+}$ is a small increment of α. We denote the samples at the (k+1)th step, which is obtained by Equation (Equation 29) as $X_{k+1,\alpha }^{⊗N}:=X_{k,\alpha }^{⊗N}-h\left(I_{dN}+\alpha J\right)\nabla {\hat{U}}^{⊗N}\;\left(X_{k,\alpha }^{⊗N}\;\right)+\sqrt{2h\beta^{-1}}\epsilon_{k}$, (or (Equation 30) as $X_{k+1,\alpha }^{⊗N}:=X_{k}^{⊗N}+\Sigma^{-1}V_{k}^{⊗N}h+\alpha J\nabla {\hat{U}}^{⊗N}\left(X_{k}^{⊗N}\right)h$). We denote the samples, which are obtained by replacing the above α by α+η, as $X_{k+1,\alpha +\eta }^{⊗N}$. We denote the KSD between the measure of $X_{k+1,\alpha }^{⊗N}$ and stationary measure π as KSD(k+1,α) and estimate the differences of empirical KSD:(34)$$\begin{array}{cc} \; & \Delta :=\hat{KSD}\left(k+1,\alpha \right)-\hat{KSD}\left(k+1,\alpha +\eta \right), \end{array}$$
where KSD is estimated by
(35)KSD^(k,α)=1N(N−1)∑i=1Nuq(Xk,α(i),Xk,α(j)),
(36)uq(x,x′):=∇xlogπ(x)⊤l(x,x′)∇xlogπ(x′)+∇xlogπ(x)⊤∇x′l(x,x′)+∇xl(x,x′)⊤∇xlogπ+Tr∇x,x′l(x,x′),
where *l* denotes a kernel and we use an RBF kernel. If Δ>0, which indicates that the empirical distribution of Xk+1,α+η⊗N is closer to the stationary distribution than that of Xk+1,α⊗N. Thus, we should increase the interaction strength from α to α+η. If Δ<0, we decrease it to α−η. We also update η to cη where c∈(0,1]. The overall process is shown in Algorithm 1. Detailed discussions of the algorithm including how to select α0,η0, and *c* are shown in Appendix L.
**Algorithm 1 **Tuning α**Input: **Xk⊗N,ηk,αk,c**Output: **αk+1,ηk+1  1:Calculate Xk+1,αk⊗N and Xk+1,αk+ηk⊗N.  2:Calculate Δ:=KSD^(k+1,αk)−KSD^(k+1,αk+ηk)  3:**if **Δ>0** then**  4: Update αk+1=αk+ηk  5: Update ηk+1=ηk  6:**else**  7: Update αk+1=|αk−ηk|  8: Update ηk+1=cηk  9:**end if**

Finally, we present Algorithm 2, which describes the whole process. We update the value of α once every k′ step. Please note that its computational cost is not much larger than that of Equation (Equation 30). We only calculate the eigenvalues of *J* once, which requires O(N3). The calculation of different KSDs is computationally inexpensive since we can re-use the gradient, which is the most computationally demanding part.
**Algorithm 2 **Proposed algorithm**Input: **X0⊗N,h,α0,η,k′,K,c,(V0⊗N,γ,Σ−1)**Output: **XK⊗N  1:Make a N×N random matrix J0 and check kerJ0={0}  2:Set J=J0⊗Id  3:**for** k=0 to *K* **do**  4: **if **
⌊kk′⌋=0
** then**  5:  Update α by Algorithm 1  6: **end if**  7: Update Xk⊗N by Equation (Equation 29) (for skew-SGLD)  8: (Update (Xk⊗N,Vk⊗N) by Equation (Equation 30) for skew-SGHMC)  9:**end for**

### 4.3. Refined Analysis for the Bias of Skew-SGLD

When using a constant step size for skew-SGLD, the bound in Theorem 7 is meaningless since the first term of Equation (Equation 32) will diverge. Here, following [23], we present a tighter bound for the bias of skew-SGLD under a stronger assumption.

**Theorem** **8.**
*Under Assumptions 1–7, for any k∈N and any $h\in (0,1\wedge \frac{\lambda (\alpha ,N)}{4\sqrt{2}M^{2}}\wedge \frac{m}{4M^{2}})$ obeying kh≥1 and βm≥2, we have*
(37)$$\begin{array}{cc} \; & \left|E\frac{1}{N}\sum_{n=1}^{N}f\left(X_{k}^{\left(n\right)}\right)-\int_{R^{d}}fd\pi \right|\leq L_{f}\sqrt{\frac{2}{\lambda (\alpha ,N)}}\sqrt{e^{-\lambda (\alpha ,N)kh}\mathrm{KL}\left(\mu_{0}|\pi \right)+\frac{C_{3}\left(\alpha \right)}{\lambda (\alpha ,N)}}, \end{array}$$
*where*
(38)$$\begin{array}{c} \lambda \left(\alpha ,N\right):={\left(\frac{1}{\left(1+m{\left(\alpha ,N\right)}^{-1}\beta C\left(m_{0}\right)\right)2\pi e^{2}}+\frac{3}{2}m{\left(\alpha ,N\right)}^{-1}\right)}^{-1} \end{array}$$
*and constants $C_{3}\left(\alpha \right)$ and $C(m_{0})$ depend on the constants of Assumptions 1–7. Moreover, λ(α,N) satisfies λ(α,N)≥λ(α=0,N). For the details, see Appendix M.*


Proof is shown in Appendix M. Please note that even if we use a constant step size for skew-SGLD, the bound in Theorem 8 will not diverge. Here we need the stronger assumption about a step size compared to Theorem 7. From Equation (Equation 37), the convergence behavior is characterized by λ(α,N) and the bias bound become smaller when λ(α,N) become larger. From the definition of λ(α,N), the larger m(α,N) is, the larger λ(α,N) we obtain. Thus, as we had seen so far, introducing the skew matrices leads to the larger Poincaré constant, and thus, this leads to larger λ(α,N).

Previous work [18] clarified that if α is sufficiently small, introducing skew matrices improves the Poincaré constant by a constant factor, which means that we have $m\left(\alpha ,N\right)-m_{0}\approx O\left(\alpha^{2}\right)$, where $O(\alpha^{2})$ depends on the eigenvector and eigenvalues of the generator L. On the other hand, from Theorem 8, for any ξ>0, to achieve the bias smaller than ξ, it suffice to run skew-SGLD at least for $k\geq \frac{2}{\lambda (\alpha ,N)h}\ln\frac{L_{f}}{\xi }\sqrt{\frac{\mathrm{KL}(\mu_{0}|\pi )}{2\lambda (\alpha ,N)}}$ iterations using the appropriate step size *h* and under the assumption that δ and α are small enough (see Section M.2 for details). Combined with these observations, introducing skew matrices into SGLD improves the computational complexity for a constant order. Our numerical experiments show that even constant improvement results in faster convergence in practical Bayesian models.

## 5. Related Work

In this section, we discuss the relationship between our method and other sampling methods.

### 5.1. Relation to Non-Reversible Methods

As we discussed in Section 1, our work extends the existing analysis of non-reversible dynamics [8,18] and presents a practical algorithm. Compared to those previous works, we focus on the practical setting of Bayesian sampling and derive the explicit condition about *J* for acceleration. We also derived a formula to quantitatively evaluate skew acceleration based on the asymptotic expansion of the eigenvalues of the perturbed Hessian matrix. A previous work [24], which derived the optimal skew matrices when the target distribution is Gaussian, requires $O(d^{3})$ computational cost to derive optimal skew matrices, and it is unclear whether it works for non-convex potential functions. On the other hand, our construction method for skew matrices is simple, computationally cheap, and can be applied to general Bayesian models.

Our work analyzes skew acceleration for ULD, which is more effective than LD in practical problems. Another work [8,18] only analyzed skew acceleration for LD. A previous work [17] combined a non-reversible drift term with ULD. Unlike our method, this work’s purpose was to reduce the asymptotic variance of the expectation of a test function and is mainly focusing on sampling from Gaussian distribution.

To the best of our knowledge, our work is the first to focus on the memory issue of skew acceleration and develop a memory-efficient skew matrix for ensemble sampling. Our work also presents an algorithm that controls the trade-off for the first time. Another work [18] identified the trade-off and handled it by cross-validation, which is computationally inefficient, unfortunately.

Finally, we point out an interesting connection between our skew-SGHMC and the magnetic HMC (M-HMC) [25]. M-HMC accelerates HMC’s mixing time by introducing a “magnetic” term into the Hamiltonian. That magnetic term is expressed by special skew matrices. Although a previous work [25] argued that M-HMC is numerically superior to a standard HMC, its theoretical property remains unclear. Thus, our work can analyze the theoretical behavior of magnetic HMC.

### 5.2. Relation to Ensemble Methods

Our proposed algorithm is based on ensemble sampling [26]. Ensemble sampling, in which multiple samples are simultaneously updated with interaction, has been attracting attention numerically and theoretically because of improvements in memory size, computational power, and parallel processing computation schemes [26]. There are successful, widely used ensemble methods, including SVGD [27] and SPOS [28], with which we compare our proposed method numerically in Section 6. Although both show numerically good performance, it is unclear how the interaction term theoretically accelerates the convergence since they are formulated as a McKean–Vlasov process, which is non-linear dynamics, complicating establishing a finite sample convergence rate. Our algorithm is an extension of another work [18], where the interaction was composed of a skew-acceleration term and can be rigorously analyzed. Compared to that previous work [18], we analyzed skew acceleration, focused on the Hessian matrix, and developed practical algorithms, as discussed in Section 4.2, and derived the explicit condition when acceleration occurs, which was unclear [18].

Another difference among SPOS, SVGD, and [18] is that they use first-order methods; our approach uses the second-order method. Little work has been done on ensemble sampling for second-order dynamics. Recently a second-order ensemble method was proposed [29], based on gradient flow analysis. Although its method showed good numerical performance, its theoretical property for finite samples remains unclear since it proposed a scheme as a finite sample approximation of the gradient flow. In contrast, our proposed method is a valid sampling scheme with a non-asymptotic guarantee.

## 6. Numerical Experiments

The purpose of our numerical experiments is to confirm the acceleration of our algorithm proposed in Section 4 in various commonly used Bayesian models including Gaussian distribution (toy data), latent Dirichlet allocation (LDA), and Bayesian neural net regression and classification (BNN). We compared our algorithm’s performance with other ensemble sampling methods: SVGD, SPOS, standard SGLD, and SGHMC. In all the experiments, the values and the error bars are the mean and the standard deviation of repeated trials. For all the experiments we set γ=1 and $\Sigma^{-1}=300$ for SGHMC and Skew-SGHMC. As for the hyperparameters of our proposed algorithm, the selection criterion is discussed in Appendix L.

### 6.1. Toy Data Experiment

The target distribution is the multivariate Gaussian distribution, π=N(μ,Ω) where we generated $\Omega^{-1}=A^{⊤}A$ and each element of $A\in R^{2d\times d}$ is drawn from the standard Gaussian distribution. The dimension of the target distribution is d=50, we approximate by 20 samples using the proposed ensemble methods. We tested these toy data because the LD for this target distribution is known as the Ornstein–Uhlenbeck process, and its theoretical properties have been studied extensively e.g., [30]. Thus, by studying the convergence behavior of these toy data, we can understand our proposed method more clearly.

First, we confirmed how the skew-symmetric matrix affects the eigenvalues of the Hessian matrix, as discussed in Section 3, where we only showed the asymptotic expansion for the smallest real part of the eigenvalues and saddle point. Here we can show a similar expansion for the largest real part: (39)ReλdNα=λdN0+α2∑k=1dN−1λdN0λk0|vk0JvdN0|2λk0−λdN0+O(α3).

ReλdNα≤λdNα holds.

Then we observed how the largest and smallest real parts of the eigenvalues of $\left(I+\alpha J\right)\Omega^{-1}$ depend on α. The results are shown in Figure 2, where we averaged 10 trials over a randomly made *J* with fixed *A*. The upper-left, upper-right, and lower figures show Re(λ1(α)), Re(λdN(α)), and Re(λ1(α))/Re(λdN(α)). These behaviors are consistent with Theorem 3. When α is small, its behavior is close to the quadratic function proved in Theorem 3.

Next, we observed the convergence behavior of skew-SGLD and skew-SGHMC. We measured the convergence by maximum mean discrepancy (MMD) [31] between the empirical and stationary distributions. For MMD, we used 2000 samples for the target distribution, and we used the Gaussian kernel whose bandwidth is set to the median distance of these 2000 samples. We used gradient descent (GD), with step size $h=1\times 10^{-4}$. The results are shown in Figure 3. The proposed method shows faster convergence than naive parallel sampling, which is consistent with Table 2.

### 6.2. LDA Experiment

We tested with an LDA model using the ICML dataset [32] following the same setting as [33]. We used 20 samples for all the methods. Minibatch size is 100. We used step size $h=5\times 10^{-4}$. First, we confirmed the effectiveness of our proposed Algorithm 1, which adaptively tunes α values. For that purpose, we compared the final performance obtained by our methods with a previous method [18], in which α is selected by cross-validation (CV). Here instead of CV, we just fixed α during the sampling and refer to it as fixed α. We also tested the case when *J* is generated randomly at each step with fixed α, as discussed in Section 4.1. We refer to it as random J. The results are shown in Figure 4 where skew-SGLD was used. We found that our method showed competitive performance with the best performance of fixed α. For the computational cost, we used $k^{\prime }=2$ in Algorithm 2, and our method needed twice the wall clock time than each fixed α. This means that our algorithm greatly reduces the total computational time since we tried more than two αs in the fixed α for CV. We also found that since using different *J*s at each step did not accelerate the performance, we need to store and fix *J* during the sampling for acceleration. Next, we compared our method with other ensemble sampling schemes and observed the convergence speed. The result is shown in Figure 5. Skew-SGLD and skew-SGHMC outperformed SGLD and SGHMC, which is consistent with our theory.

### 6.3. BNN Regression and Classification

We tested with the BNN regression task using the UCI dataset [34], following a previous setting Liu and Wang [27]. We used one hidden layer neural network model with ReLU activation and 100 hidden units. We used 10 samples for all the methods. We used the minibatch size 100. We used step size $h=5\times 10^{-5}$. The results are shown in Table 1 and Table 2. We also tested on BNN classification task using the MNIST dataset. The result is shown in Figure 6. We used one hidden layer neural network model with ReLU activation and 100 hidden units. Batchsize is 500 and we set step size $h=5\times 10^{-5}$. Our proposed methods outperformed other ensemble methods. Please note that skew-SGHMC and skew-SGLD consistently outperformed SGHMC and SGLD.

## 7. Conclusions

We studied skew acceleration for LD and ULD from practical viewpoints and concluded that the improved eigenvalues of the perturbed Hessian matrix caused acceleration and derived the explicit condition for acceleration. We described a novel ensemble sampling method, which couples multiple SGLD or SGHMC with memory-efficient skew matrices. We also proposed a practical algorithm that controls the trade-off of faster convergence and larger discretization and stochastic gradient error and numerically confirmed the effectiveness of our proposed algorithm.

## Figures and Tables

**Figure 1 entropy-23-00993-f001:**
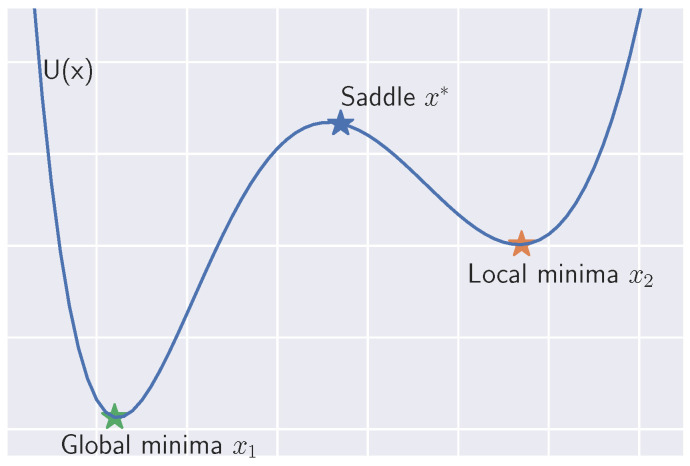
Double-potential example: Poincaré constant is related to the eigenvalue at $x^{*}$.

**Figure 2 entropy-23-00993-f002:**
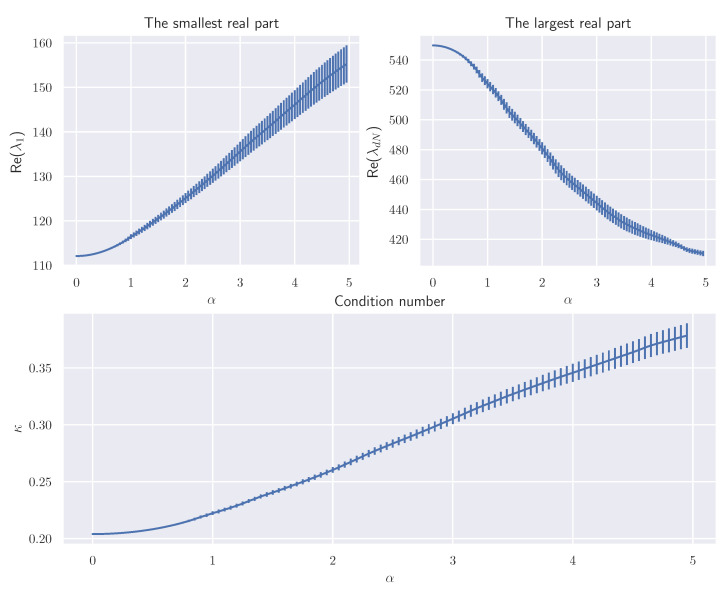
Eigenvalue changes (averaged over ten trials).

**Figure 3 entropy-23-00993-f003:**
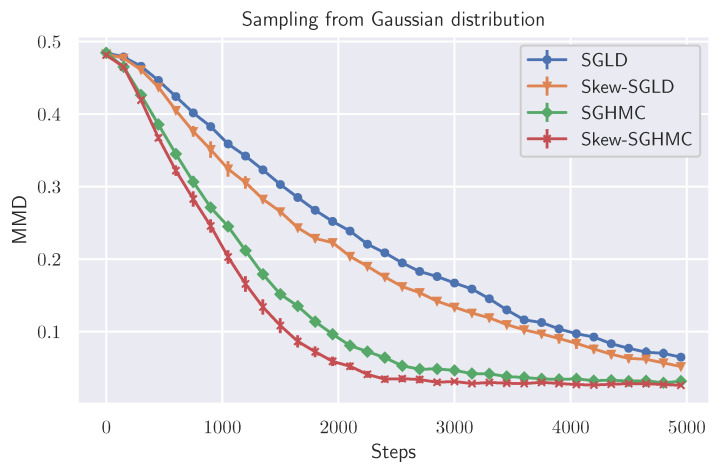
Convergence behavior of toy data in MMD (averaged over ten trials).

**Figure 4 entropy-23-00993-f004:**
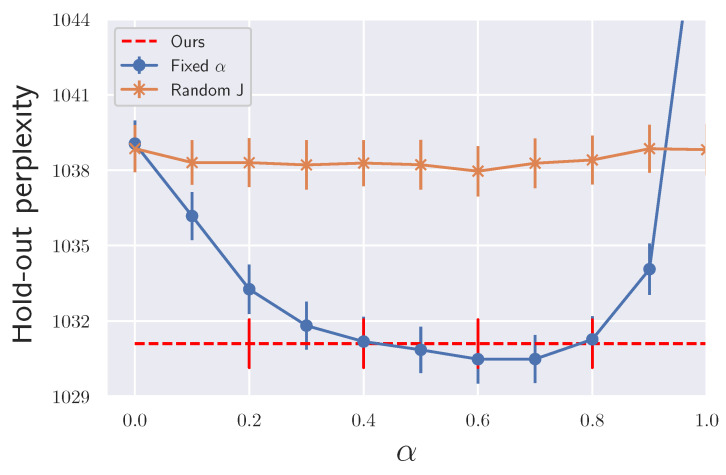
Final performances of LDA under different values of α (averaged over ten trials).

**Figure 5 entropy-23-00993-f005:**
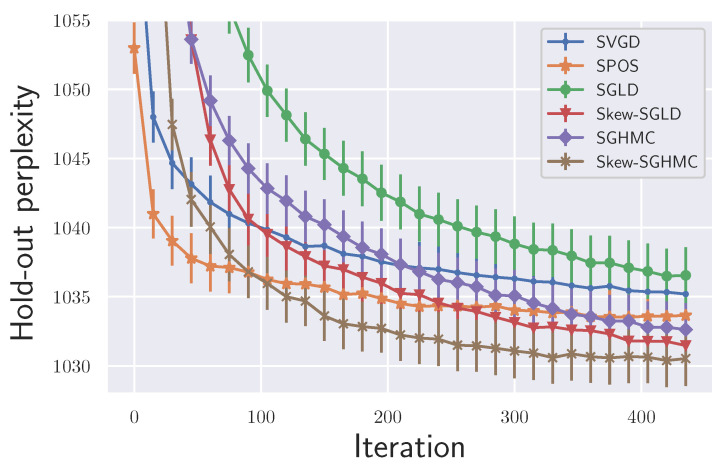
LDA experiments (Averaged over 10 trials).

**Figure 6 entropy-23-00993-f006:**
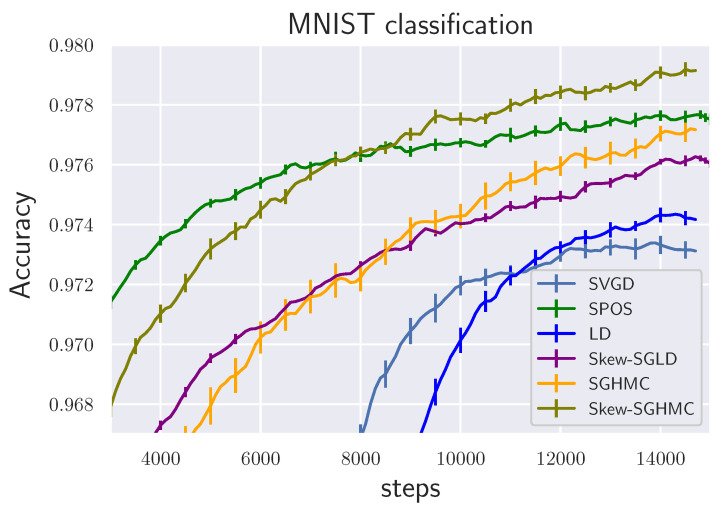
MNIST classification (Averaged over ten trials).

**Table 1 entropy-23-00993-t001:** Benchmark results on test RMSE for regression task.

Dataset	Avg. Test RMSE					
	SVGD	SPOS	SGLD	Skew-SGLD	SGHMC	Skew-SGHMC
Concrete	5.709 ± 0.040	5.239 ± 0.199	5.009 ± 0.091	4.973 ± 0.057	4.949 ± 0.144	4.790 ± 0.081
Kin8nm	0.0731 ± 0.0006	0.0688 ± 0.0003	0.0693 ± 0.0006	0.0689 ± 0.0005	0.0687 ± 0.0001	0.0683 ± 0.0003
Energy	0.520 ± 0.060	0.456 ± 0.030	0.428 ± 0.045	0.412 ± 0.045	0.406 ± 0.019	0.403 ± 0.008
Bostonhousing	3.306 ± 0.005	3.107 ± 0.173	2.948 ± 0.084	2.930 ± 0.095	3.053 ± 0.093	2.986 ± 0.143
Winequality	0.619 ± 0.001	0.618 ± 0.007	0.641 ± 0.003	0.634 ± 0.004	0.614 ± 0.004	0.613 ± 0.004
PowerPlant	4.219 ± 0.012	4.160 ± 0.009	4.129 ± 0.002	4.118 ± 0.006	4.112 ± 0.009	4.105 ± 0.008
Yacht	0.475 ± 0.049	0.467 ± 0.110	0.464 ± 0.058	0.442 ± 0.046	0.464 ± 0.078	0.432 ± 0.051

**Table 2 entropy-23-00993-t002:** Benchmark results on test negative log likelihood for regression task.

Dataset	Avg. Test Negative Log Likelihood					
	SVGD	SPOS	SGLD	Skew-SGLD	SGHMC	Skew-SGHMC
Concrete	−3.157 ± 0.008	−3.124 ± 0.025	−3.052 ± 0.009	−3.049 ± 0.012	−3.046 ± 0.025	−3.033 ± 0.021
Kin8nm	1.153 ± 0.0084	1.212 ± 0.008	1.223 ± 0.002	1.223 ± 0.005	1.230 ± 0.0015	1.235 ± 0.0025
Energy	−0.816 ± 0.102	−0.976 ± 0.079	−0.867 ± 0.056	−0.845 ± 0.021	−0.843 ± 0.045	−0.844 ± 0.041
Bostonhousing	−2.98 ± 0.000	−2.644 ± 0.027	−2.548 ± 0.016	−2.539 ± 0.002	−2.574 ± 0.019	−2.561 ± 0.017
Winequality	−1.012 ± 0.000	−0.959 ± 0.007	−0.976 ± 0.006	−0.968 ± 0.005	−0.941 ± 0.007	−0.938 ± 0.005
PowerPlant	−2.871 ± 0.004	−2.850 ± 0.004	−2.844 ± 0.002	−2.842 ± 0.001	−2.838 ± 0.004	−2.835 ± 0.003
Yacht	−1.184 ± 0.06	−1.372 ± 0.07	−1.077 ± 0.066	−1.078 ± 0.030	−1.083 ± 0.030	−1.079 ± 0.051

## Data Availability

Publicly available datasets were analyzed in this study. This data can be found here: http://archive.ics.uci.edu/ml (accessed on 21 June 2021).

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
