# Peer review of "Accelerated Diffusion-Based Sampling by the Non-Reversible Dynamics with Skew-Symmetric Matrices"

_entropy, 2021, doi:10.3390/e23080993_

Round 1
Reviewer 1 Report
See attached

Reviewer 2 Report
The authors articulate the problems in the earlier literature and have convincingly addressed the problem. The lack of theoretical underpinnings in the literature have been dealt by the authors carefully and made a significant contribution to the literature. The authors contribution to the literature is two fold: (i) they present a convergence analysis of skew acceleration for standard Bayesian model settings, including non-convex potential functions using Poincare constants, and (ii) they develop a practical skew accelerated sampling algorithm for a parallel sampling setting with a memory-efficient skew matrices.
The results are supported by their simulation studies.
My minor comment is: it would be interesting to see the practical implications of their application with appropriate policy implications, rather than simply applying to a data set.
Author Response
Thank you very much for your valuable comments on our work.
Based on your comment about the practical implications of the proposed method, we add the following explanation about ensemble sampling in Section 5 in the revised paper,
“Ensemble sampling, in which multiple samples are simultaneously updated with interaction, has been attracting attention numerically and theoretically because of improvements in memory size, computational power, and parallel processing computation schemes”.
Please see the revised paper for other changes.
Reviewer 3 Report
The manuscript deals with the comparison of the performance of several algorithms for solving the Langevin dynamics equation for a general system. For this purpose, the authors develop a convergence analysis of skew acceleration based on a Hessian matrix and Pointaré constants. This allows establishing a clear criterion to select an algorithm for a particular system. In addition, the authors present a new skew acceleration for underdamped Langevin dynamics, which shows a fast convergence while keeping a low memory demand. I think these findings deserve publication. Besides, the paper is well-written. Important results are shown in the main body, and the proofs are given in the appendixes. My only concern is the journal selection. I am not sure the ms fits the scopes of Entropy. In particular, I find its focus closer to a mathematical or a computational-specialized journal, such as the several appearing as references. Nonetheless, I prefer to let the editor judge in this regard.
I have few comments and found few details the authors may consider. These are:
It says "Although ULD has been widely used in practice, the application of skew acceleration is limited although they are expected to show superior performance theoretically." - > it (the application) is expected. This sentence appears in the abstract and the introduction.
It says "In MCMC, Langevin dynamics (LD) is a popular choice because of its superior theoretical and numerical performance." - > This is not why LD is popular. LD accounts for omitted degrees of freedom by making use of stochastic differential equations. For instance, it has no sense to use LD when explicitly accounting for the solvent. Thus, LD is frequently used to deal with colloidal dispersions. LD has the disadvantage of making it necessary to handle hydrodynamic interactions. I think you should explain this in the introduction.
I think you are restricting the study to bulk systems with no external field containing only a single species (a unique u(x) function), but I am not sure (may the set x account for different species?). In any case, you should make this clearer.
Poincare - > Poincaré
\beta is not defined in Eq. (1).
It says "The stationary distribution of S-LD is still π" - > a ~ is missing above π.
It says "whose stationary distribution is π̃:= ... " - > Since π̃ was previously defined you do not need to write the right-hand side of this equality.
It says "For example, skew acceleration has been analyzed focusing on sampling from Gaussian distributions [13–17], although such an assumption is not always valid in practice." - > Which assumption? The Gaussian distribution? Please, rephrase.
It says "we use the random matrix property:" - > something seems to be missing after :
We denote the these smallest - > delete the
